# Differential Diagnosis of Visual Phenomena Associated with Migraine: Spotlight on Aura and Visual Snow Syndrome

**DOI:** 10.3390/diagnostics13020252

**Published:** 2023-01-09

**Authors:** Eliseo Barral, Elisa Martins Silva, David García-Azorín, Michele Viana, Francesca Puledda

**Affiliations:** 1Department of Neurology, Sanatorio Allende Cerro, Córdoba X5021FPQ, Argentina; 2Department of Neurology, Hospital Garcia de Orta, 2805-267 Almada, Portugal; 3Headache Unit, Department of Neurology, Hospital Clínico Universitario de Valladolid, Gerencia Regional de Salud de Castilla y Leon (SACYL), 47003 Valladolid, Spain; 4Headache Group, Wolfson CARD, Institute of Psychiatry, Psychology & Neuroscience, King’s College London, London SE5 9PJ, UK

**Keywords:** migraine, aura, visual snow syndrome, headache, photophobia

## Abstract

Migraine is a severe and common primary headache disorder, characterized by pain as well as a plethora of non-painful symptoms. Among these, visual phenomena have long been known to be associated with migraine, to the point where they can constitute a hallmark of the disease itself. In this review we focus on two key visual disorders that are directly or indirectly connected to migraine: visual aura and visual snow syndrome (VSS). Visual aura is characterized by the transient presence of positive and negative visual symptoms, before, during or outside of a migraine attack. VSS is a novel stand-alone phenomenon which has been shown to be comorbid with migraine. We discuss key clinical features of the two disorders, including pathophysiological mechanisms, their differential diagnoses and best treatment practices. Our aim is to provide an aid for clinicians and researchers in recognizing these common visual phenomena, which can even appear simultaneously in patients with an underlying migraine biology.

## 1. Introduction

Migraine is a prevalent neurological disorder, listed by the Global Burden of Disease Study as the second leading cause of years lived with disability worldwide [1]. It affects around 15% of the general population globally and is typically characterised by recurring, highly disabling attacks of severe head pain lasting for 4–72 h, associated with nausea, vomiting, increased sensitivity to light and sound and numerous neurological signs and symptoms [2]. The diagnosis of migraine is clinical and based on criteria provided in the third edition of the International Classification of Headache Disorders (ICHD-3) [3]. Migraine being a primary headache, individuals will typically have a normal physical examination with no underlying causes for their pain [3,4]. However, migraine is much more than a disorder of head pain, and alteration of vision as a neurological dysfunction symbolises one of the hallmarks of the disease. In fact, up to a third of patients may experience transient focal neurological symptoms in at least some of their attacks which fulfil the diagnostic criteria for the condition of migraine with aura (MwA) [5].

The term aura derives from the ancient Greek *αύρα*, meaning ‘breeze, soft wind’, and is defined as recurrent attacks, lasting minutes, of unilateral fully-reversible visual, sensory or other central nervous system symptoms that usually develop gradually and are typically followed by migraine headache [3,6]. Visual aura symptoms (VASs) are by far the most common presentations and occur in 98–99% of cases of aura. Other symptoms such as disturbances of sensation and language occur respectively in 36% and 10% of aura episodes [7]. Hemiplegic aura is much rarer, with a reported prevalence of 0.01% in the general population [8].

Visual disorders are also a common feature of migraine outside of the context of aura. Some patients, for example, can experience transient visual disturbances during the migraine attack lasting only a few seconds and over the bilateral visual field [9]. Further, some patients with migraine biology seem to be predominantly affected by the migraine-associated phenomenon of visual snow [10].

This narrative review has the purpose of examining the main clinical features of typical and atypical visual aura, the newly described phenomenon of visual snow and associated syndrome, and to aid the clinician in the differential diagnosis between these disorders. A brief overview is also given of known pathophysiological mechanisms of these two distinct but related conditions, as well as current treatment options.

## 2. Migraine Aura

### 2.1. Clinical Features of Migraine Aura

Migraine aura is both an alarming symptom to patients and an intriguing phenomenon to clinicians and scientists. Approaching a patient with MwA in a clinical setting can be challenging, due to the heterogeneity of the condition and the complexity of the patient’s history. An accurate clinical characterisation of aura symptomatology can improve the diagnostic accuracy and identification of patients that suffer from this condition, particularly since there are no biological markers of aura that can be used in routine clinical practice.

History taking of VASs should focus on all the following symptom features: frequency, consistency, uniformity, duration, timing, temporal relation to the headache, characteristics of the visual phenomena, location in the visual field, laterality (one eye/both eyes) progression (modification of visual disturbance), direction of aura spreading in the visual field. Importance should also be given to features of previous headache attacks, in order to establish a diagnosis of episodic or chronic migraine and the presence of non-visual aura symptoms [5]. There is high variability in the frequency of aura presentation, as some patients can experience a single visual aura episode in the context of a lifetime history of frequent migraines or have all of their migraine attacks accompanied by visual symptoms. The patient’s history typically reflects the true perception of what they see, and malingering or psychogenic causes should be considered the exception when detailing visual symptoms in migraineurs [6].

VASs are pleomorphic and clinical studies have shown that patients can report a plethora of different, often complex, visual disturbances that may include positive visual phenomena, negative visual phenomena, and/or disturbances of visual perception. These visual phenomena could effectively be defined by subdividing the perceived visual scenarios into so-called elementary visual symptoms (EVS—see Table 1 [7,11,12,13]). The ‘classic’ visual aura usually refers to: fortification spectra, which look like a fortified town as viewed from above, or teichopsia (‘seeing fortifications’), which is a jagged figure with fortification lines arranged at right angles to one another beginning from a paracentral area which usually spreads outwards leaving visual loss behind. There are often scintillations which may be white, grey, or have colours similar to a kaleidoscope in a semicircle or C shape surrounding the scotoma or area of visual loss. Scintillating scotomas are typically in one hemifield with visual field defects beginning around fixation and spreading outward [14]. Although teichopsia is considered the typical visual aura symptom, the most prevalent EVSs found in different studies have actually been dots or flashing lights [15,16,17]. It is interesting to note that a significant number of patients report more than one type of EVS in various combinations, and that VASs can have a biphasic quality, with positive phenomena (e.g., shimmering lights, or zig-zag visual disturbances) appearing first, followed within a few minutes by negative symptoms (e.g., scotoma or loss of visual image) [18].

**Table 1 diagnostics-13-00252-t001:** Common elementary visual symptoms in migraine with visual aura. Adapted from Viana et al. [7].

Positive visual phenomena Bright light Small bright dots White dots/round forms Coloured dots/round forms Lines (coloured lines) Geometrical shapes Fortification spectra (teichopsia—further detailed in the text) Zigzag or jagged lines
Negative visual phenomena Scotoma (single blind area) Scotomata (several blind/black areas) Hemianopsia (blindness in half of the visual field)
Disturbances of visual perception Visual snow (dynamic, continuous, tiny dots (see Table 2) Foggy or blurred vision Looking through heat waves/water/oil Tunnel vision (blindness in the whole periphery) Mosaic-like vision Fractured vision Micropsia (objects appear smaller than they actually are) Macropsia (objects appear larger than they actually are) Pelopsia (objects appear closer than they actually are) Telopsia (objects appear further away than they actually are) Alice in Wonderland Syndrome (a combination of micro/macro/pelo/telopsia, metamorphopsia and dissociation) Corona phenomena (i.e., an extra edge on objects) Complex hallucination (i.e., visual perception of something not present such as objects, animals, and persons) Like the negative of a film

**Table 2 diagnostics-13-00252-t002:** Symptoms and comorbidities of visual snow syndrome.

Symptom	Description	Frequency	Notes and Differential Diagnoses
Visual Snow	Uncountable tiny dots in the entire visual field, unremitting for several months Usually black/grey on white background and grey/white on black background; They can also be transparent, white flashing or coloured	Main symptom (100%)	Without other symptoms and normal evaluation, consider primary visual snow
Palinopsia	Continued perception of an object after it is no longer in the visual field: - Afterimages (indistinct, isochromatic shapes) - visual trailing (ghost images of the original moving source)	33–85%	Consider retinal afterimages, if they only occur when staring at a high contrast image and are in complementary colour Exclude cortical lesions of the posterior visual pathway [19]
Enhanced entopic phenomena	- Excessive floaters in both eyes - Excessive blue field entoptic phenomenon (Scheerer phenomenon); uncountable little grey/white/black dots or rings shooting over visual field in both eyes when looking at homogeneous bright surfaces (e.g., blue sky) - Spontaneous photopsia—bright flashes of light - Self-light of the eye—coloured waves, swirls or clouds when closing the eyes in the dark	71–86% 41–76% 44–71% 53–57%	Exclude ophthalmic pathology
Nyctalopia	Poorer vision in dim light	44–78%	
Photophobia	Continuous hypersensitivity to light	54–81%	Independent from comorbid migraine or aura Can have similar range of a migraine attack [20]
Comorbid conditions			
Tinnitus		34–75%	Overlap with other perceptual disorders, including similar pathophysiology and frequent migraine comorbidity [21] Tinnitus and migraine may worsen VSS
Migraine		52–72%	
Other neurological comorbidities	- Paresthesia - Tremor - Dizziness and balance problems - Postural orthostatic tachycardia syndrome - Fibromyalgia - Difficulty concentrating, irritability and lethargy	/	
Psychiatric comorbidities	- Anxiety - Depression - Derealization	frequent	Treatment should be offered

In their seminal study, Hansen et al. [22] analysed more than 1000 visual auras of an individual patient over almost 18 years, and the most frequent time courses of the visual symptoms were consistent with the classic visual aura starting in the centre of the visual field and then spreading centrifugally in one hemifield over about 30–60 min. Nevertheless, a considerable amount of auras were found to be different, either starting from the periphery or disappearing from the visual field and then reappearing at a distant location. Regarding aura duration, the ICHD-3 establishes that migraine aura must be considered to be typical when lasting between five and sixty minutes, and it also labels aura lasting longer than an hour and less than a week as probable migraine with aura [3]. The term ‘probable’ used in such classification indicates suspicion as to whether the symptom is in fact migraine aura. However, Viana et al. demonstrated that auras lasting more than sixty minutes are not unusual. In this study, that included 216 aura episodes, prolonged auras represented 17% of all auras and occurred at least once in 26% of patients [23]. Additionally, recent studies have also found that around 5% of aura symptoms last over 4 h, with some authors starting to consider this as a possible cut-off time for ‘typical’ aura duration [24,25]. Regarding the temporal relationship between headache and aura, although aura has been traditionally described as preceding the headache, recent data shows that these phases can actually overlap. The headache phase may in fact start before, simultaneously with the onset of aura, during the aura, simultaneously with the end, or well after the end of the aura [24,25]. Further, it is not rare for aura to occur in the absence of headache, making diagnosis more challenging.

### 2.2. Pathophysiology of Aura

Although the underlying mechanism of aura is still under open debate, indirect evidence supports the widely accepted hypothesis of aura being caused by cortical spreading depression (CSD) [26]. CSD, initially described by Leao in 1944 [27], is a bioelectrical phenomenon consisting of a wave of intense cortical neuronal activity associated with hyperemia, followed by a more prolonged period of neuronal activity suppression associated with cortical oligemia [28,29]. While in humans CSD has not been directly demonstrated during aura, the correlation between the neurophysiological characteristics of the phenomenon, its retinotopic propagation over the visual cortex, and the characteristics and dynamics of the visual deficits, seem to confirm CSD as the pathophysiological correlate of aura [25]. In fact, some authors consider that the term ‘typical migraine aura’ should be reserved for symptoms that are consistent with CSD, irrespective of the occurrence being prior, together, or independently from headache attacks [6].

However, the reasons why the visual disturbances prevail in migraine compared with other sensory modalities are not explained by CSD, and still remain poorly understood. Some authors have hypothesized that although aura might start in multiple sites of the visual cortex in the same individual, certain areas show higher propensity to be the initiating focus and even that aura may propagate silently in the cortex, without clinical manifestations [22].

### 2.3. Differential Diagnosis of Aura

Although migraine is arguably the most common cause of concurrent neurologic symptoms and headache, it is not the only setting in which this occurs. Care should be taken in the evaluation of neurologic symptoms of migraine in order to exclude serious and potentially life-threatening conditions, such as cerebrovascular disease, epilepsy, idiopathic intracranial hypertension (IIH), and psychiatric disorders (Figure 1). This is particularly the case when aura is not followed by migraine headache, when it occurs for the first time after age 40, when symptoms are exclusively negative (e.g., hemianopia) or progress very quickly, or when the episode is either very long or very short.

Migraine is the third most common stroke mimic, following seizures and psychiatric disorders, and accounts for 18% of all improper thrombolytic treatment [7]. Conversely, patients with overlooked strokes in an emergency department setting most often receive an initial misdiagnosis of migraine [30]. Transient ischemic attacks in particular can be difficult to rule out in patients in which the onset of aura symptoms is abrupt or they have an atypical duration [31]. Strokes rarely cause positive visual symptoms, but it is important to remember disorders such as Charles Bonnet syndrome, characterized by visual hallucinations in the presence of severe visual impairment, and which may follow an occipital stroke with visual field disturbance, particularly in the elderly [32]. Further visual symptoms, most commonly seen as stroke signs such as prosopagnosia, visual agnosia and hemianopia, may also occur in migraine as VASs [33]. However, stroke should be considered as the first presumed diagnosis in these cases. [32].

When presenting in childhood, visual aura may be confused with the much rarer occipital epilepsy. Attacks of occipital epilepsy are characterised by their short duration (seconds), rapid onset and offset, and presence of autonomic symptoms such as vomiting, pallor, and sweating. In certain instances, it may be difficult to differentiate these from migraine aura, particularly as the pathognomonic visual symptoms of aura can be brief or absent when occurring in childhood [28].

A previously held concept was that visual auras are usually black-and-white, and occipital seizures are commonly multicoloured [24,28]. However, a high percentage of patients with migraine aura can experience colourful visual sensations, refuting the notion that the presence or absence of colour during the visual hallucinations can be fully specific in clinically distinguishing these entities [5].

Sometimes complex visual hallucinations are associated with recurrent attacks of impairment of time sense, body image, and visual analysis of the environment. In these cases, the same differential diagnoses apply as for elementary visual hallucinations, including Alice in Wonderland syndrome [12], psychiatric illness, neurodegeneration, encephalitis, and withdrawal states such as delirium tremens [5]. Finally, in the presence of visual symptoms exclusively affecting one eye, retinal and optic nerve disorders should be investigated.

Based on these often-challenging differential diagnoses, brain magnetic resonance imaging should be requested in all forms of MwA that present one or more atypical features, and this should be accompanied by electroencephalography in the case when epilepsy is suspected and/or by a full ophthalmological examination when symptoms are fully lateralized. Further, with negative symptoms such as transient visual obscuration and blurring, secondary causes of headache such as IIH should be excluded [34].

## 3. Visual Snow Syndrome

### 3.1. History and Clinical Features of Visual Snow Syndrome

Visual snow (VS) is an entity first described in a case series in 1995 by Liu et [35], as an ‘unusual complication of migraine’, while the term used to describe it was introduced a decade later [36]. The clinical criteria of visual snow syndrome (VSS) were defined by Schankin and Goadsby [37,38] and later included in the appendix of the ICHD-3. The main clinical features of visual snow syndrome are summarized in Table 2.

Being an underrecognized disorder, VSS has often been misdiagnosed, with possible attributions varying from functional neurological disease to persistent aura. The stereotypic descriptions of the syndrome and the recent research on its pathophysiology, however, show unique brain changes leading to a distinctive dysfunctional central sensory processing [39]. The high comorbidity with migraine [40], as well as aura, further suggests a shared pathophysiology among these disorders [37]. Some authors have also suggested VSS may only represent an ‘exceptionally heightened awareness’ of physiological visual phenomena [41], but this view disregards the burden and efforts to manage this condition [42]. In fact, visual snow syndrome (VSS) presents with varying levels of disability and may strongly impact the quality of life of its sufferers [43].

The ‘snow’ or ‘static’ in VS is described as a persistent and dynamic visual disturbance, characterized by numerous tiny flickering dots, similar to a detuned television. These are more frequently black and white (black/grey on a white background or grey/white on a black background) but can also be transparent or coloured. Although the snow is typically continuous [44], it can infrequently occur in an episodic fashion, either during migraine attacks or before the onset of a classic VSS form [45,46,47]. Episodes described in the literature range from a minimum of two minutes to the entire duration of a migraine attack. Prospective symptom recording on a diary in over one-hundred patients with VSS showed that visual static is quite steady over time [48]. A prospective longitudinal study conducted on seventy-eight VSS patients showed that the condition did not change for nearly a decade in most cases (80%) and no patient, over a mean of 7 years, went into spontaneous complete remission [44].

In the full visual snow syndrome, other visual and non-visual symptoms are present, making the condition more severe [43]. The typical additional symptoms are as follows: palinopsia, enhanced entoptic phenomena (which include excessive floaters in both eyes, excessive blue field entoptic phenomenon, self-lighting of the eye and spontaneous photopsia), photophobia, and nyctalopia. Palinopsia is characterized by positive afterimages that can be illusory or hallucinatory [19]. The latter are very realistic, previously seen images, occurring with no external stimulus, frequently with concomitant visual field defects, and should warrant caution as they can be related to structural disease. In illusory palinopsia the primary image suffers a distortion, it is not as clear as the first real image and is more influenced by ambient conditions. [49] This falls into the dysfunction of visual perception which occurs in VSS. Due to symptoms such as nyctalopia and photophobia, VSS can largely impact daily life activities, such as driving at night, reading, and using screens [50]. Further symptoms that have previously been reported but that are not included in the definition of VSS are: ‘straight lines moving across the visual field’, ‘water running down a window’ and kaleidoscopes of colours even with eyes open [38]. Some authors have proposed inclusion of these in the clinical criteria, as ‘other persistent positive visual phenomena’ [51].

With regards to demographics, a recent population-based study showed the prevalence of VSS to be between 1.4% and 3.3% [41]. This study included participants unaware of the specific topic of investigation and who did not seek medical attention for their symptoms, therefore implying that a proportion of this population was not affected by symptoms in their day-to-day life [52]. VSS also shows phenotypic homogeneity across geographical areas [53].

The disorder usually starts in early life, and there seems to be no gender imbalance, much unlike migraine [43]. Nevertheless, women report higher intensity of VSS symptoms [50]. Approximately 40% of patients report symptoms for as long as they can remember, and triggers for symptom onset are often unidentifiable [43,53]. In a recent retrospective case series, however, around 40% of patients developed VS. abruptly after a perceived inciting event. Traumatic brain injury (TBI) was the most frequent of these, with VS occurring in concomitance with daily headaches and post-concussive symptoms [45]. With no optimal therapeutic approach currently available, primary cases of visual snow tend to remain stable over time.

Environmental factors that worsen visual snow are manifold, including low-light conditions, harsh artificial light, bright sunlight, and darkness [50]. Indoor and fluorescent lights may have a worse effect on symptoms when compared with natural outdoor lighting [48]. Frequently reported individual factors related to worsening include the following: fatigue, anxiety, alcohol consumption, inadequate sleep, exercise, caffeine, and screen use [45,50].

Aside from the frequent association with migraine, VSS patients often experience the non-visual sensory symptom of tinnitus (up to 70% of VSS cases [43]), usually described as high-pitched and continuous. Both migraine and tinnitus have been independently associated with a more severe presentation of VSS [43]. Patients might also experience somatosensory symptoms, or even tremor and balance problems [54]. Psychiatric comorbidity—with symptoms such as anxiety, depression, depersonalization, fatigue, poor sleep—is also common and further impacts quality of life. Depersonalisation, in particular, relates to increased severity of VSS symptoms and seems to be an intrinsic characteristic of the disorder rather than secondary to the disability caused by the symptoms [50].

### 3.2. Pathophysiology of VSS

Research over the last years has been shedding light on the mechanisms underlying VSS [55].

With regards to neuroimaging, a study using [^18^F]-FDG PET to investigate visual cortex metabolism in VSS, showed significant hypermetabolism in the lingual gyrus as well as a trend of increased metabolism in the left cerebellum [40]. The lingual gyrus, corresponding to Brodmann area 19, is part of the supplementary visual cortex, and is also involved in changes related to photophobia [56] and aura [57] in migraineurs, highlighting the underlying shared biology of the conditions.

A recent case report further showed hypoperfusion of bilateral occipital cortices in a patient with VSS and comorbid migraine with aura. Since migraine attacks had not occurred for 2 years, the authors argued that this observation in the ventral visual association stream was likely due to VSS alone [58].

fMRI studies on VSS patients also showed altered functional connectivity within the visual system, specifically the pre-cortical and cortical visual pathway, the visual motion network, the attentional and salience networks. These changes could indeed be causing a disruption in the filtering and integration of incoming sensory visual stimuli, thus explaining the percept of VSS [59]. Puledda et al. recently investigated regional cerebral blood flow, both at rest and during visual stimuli simulating visual snow, and found a specific pattern of increased regional perfusion in several brain areas which are mostly involved in complex sensory processing, such as the precuneus, supplementary motor cortex, posterior cingulate cortex, and cerebellum [60]. This data reinforces the idea of an abnormal focus on normal sensory phenomena of the VSS brain. The cerebellum, in particular, may have a key role in the biology of the condition, possibly leading to dysfunctional feed-forward mechanisms of sensory processing.

Structural brain changes have also been reported in VSS. Using voxel-based morphometry and guided by PET changes, a recent study showed increased grey matter volume in the temporal and limbic lobes, and decreased volume in the superior temporal gyrus. This finding introduced the concept that the disorder of VSS extends well beyond the visual system [61]. Another study recently confirmed this, by showing grey matter volume changes in the left occipital cortex and cerebellum in patients with VSS matched to healthy controls [62].

Electrophysiological studies support these and other changes found through neuroimaging [63], with one study showing differences attributed to abnormal processing in the extrastriate visual cortex in VSS [64]. Behavioural measures have also been consistent with visual cortical hyperexcitability in VSS [65]. Specifically, changes in visual processing manifest with significantly faster eye movements toward a suddenly appearing visual stimulus and difficulty in inhibiting eye movement toward a non-target visual stimulation [66]. Further studies have shown that a shift of attention elicits a stronger increase in saccade-related activity of patients than healthy controls, thus suggesting poor attentional control of vision in VSS [67].

### 3.3. Differential Diagnosis of VSS

Diagnosis of VSS requires a careful neuro-ophthalmic history and a thorough investigation of the visual pathway. It has in fact been shown that, although the syndrome is generally benign and primary, it may occasionally present as secondary to other causes (see Table 3) [45]. Identifying an inciting event is relevant since these are potentially treatable and tend to have better prognosis compared to spontaneously occurring VSS. Hang et al. suggest some red flags when evaluating possible VSS, including: new-onset visual snow, onset at an older age, intermittent or sudden exacerbation of VS, unilateral or hemifield VS, absence of additional visual disturbances, history of recently discontinued illicit drugs, and additional visual or neurologic changes [68].

Different ancillary exams can be used to exclude secondary causes, but there are no standard guidelines. Vaphiades et al. suggest that for those patients reporting a typical VSS history with normal neuro-ophthalmologic examination, including automated perimetry, other ancillary testing may be unnecessary [69]. Sampatakakis et al. recommend a brain MRI and EEG in all VSS patients with palinopsia, the most common additional symptom, which, as described above, can be associated with potentially severe conditions [52]. The differential diagnosis of visual snow, particularly when onset is rapid, should include bilateral optic neuropathies, such as metanol intoxication, ischemia, Leber optic neuropathy, and folate or B12 deficiency.

Hallucinogen persisting perception disorder (HPPD) can manifest in the visual snow spectrum [43]. This condition is described in the Diagnostic and Statistical Manual of Mental Disorders, Fifth Edition [70] as a re-experiencing of perceptual symptoms, usually visual, and felt during intoxication with a hallucinogenic drug. In HPPD the visual symptoms, which often resemble visual snow, can be unremitting; they can occur with different recreational drugs, including cannabis, ecstasy, lysergic acid diethylamide, and psylocibin [71]. Van Dongen and colleagues studied the association between VSS with migraine and VSS with HPPD, showing that these are phenotypically distinct populations [72]. They hypothesized that HPPD had a different initiation mechanism with respect to ‘primary’ VSS. Both conditions, however, lack good therapeutic options [71].

Finally, as detailed in Table 1, visual snow can be present episodically among the classic visual disturbances of aura, so particular attention should be made in patients who have a background of migraine and a history of visual and non-visual aura in order to appropriately distinguish the two conditions (Figure 2). A diagnostic algorithm for visual disturbances associated with migraine is shown in Figure 3.

## 4. Treatment

The majority of studies addressing the efficacy of different migraine treatments have included mixed populations of patients with or without aura, while treatment effect based on aura has been rarely reported. Under these circumstances, the current guidelines for migraine do not establish any treatment distinction regarding patients with or without visual aura [4,73,74]. However, some studies also indicate that MwA might respond differently to acute or preventive therapies as compared to migraine without aura. For instance, Hansen et al. found that sumatriptan can be less effective as acute treatment for attacks with aura compared to those without [75]. Regarding the preventive treatment for MwA, a systematic review observed that there are no trials directly comparing drug effects in patients with and without aura [67]. As CSD plays a primary role in aura pathophysiology, it could be expected that patients with MwA are more likely to respond to preventive treatments that supress or inhibit CSD. It has been shown on animal models that widely prescribed migraine preventives such as topiramate, valproate, propranolol, amitriptyline, and methysergide can in fact suppress CSD by 40% to 80%. In addition, a recent review suggests that topiramate, verapamil, magnesium and valproate should be considered as preventive medication in patients with frequent migraine aura [76]. The use of lamotrigine in migraine with aura was also recently reviewed, with authors stating that this drug could be considered as a preventive medication for MwA at a target dose of 100 mg daily if patients have failed other medications [77].

Regarding VSS, a first measure is for patients to be counselled on the benign and relatively stable nature of the condition. Pharmacologic treatment has shown little benefit, with even a possibility of worsening of symptoms which should be taken into consideration. Response to medication seems to be inferior in patients who have an earlier age of onset [48].

A case report of total remission exists after lamotrigine treatment [78], and a literature review showed the best pharmacological evidence for lamotrigine and topiramate, which were effective in 22% and 15% of patients respectively [79]. On a recent large treatment questionnaire, however, both lamotrigine and topiramate resulted in worsening (up to 35%), which was higher than their frequency of improvement (21% and 18%, respectively) [48]. This study also showed worsening with recreational drugs and alcohol intake, as well as atypical antidepressants and attention deficit hyperactivity disorder medication, which should thus be used with caution in VSS. On the other hand, vitamins and nutraceuticals represent viable options for patients with severe symptoms, given their safety profile and occasional benefit (15%). Benzodiazepines seem to have a small net benefit, which needs to be further investigated [48].

Nonpharmacological approaches such as using color filters of the yellow–blue color spectrum have shown encouraging results, with improvement around 90% on visual static in small cohorts [54,79]. Patients also report wearing sunglasses to reduce light sensitivity, improvement with healthy sleep and diet habits, and learning how to ignore symptoms [50]. Repetitive transcranial magnetic stimulation (rTMS) as a potential treatment for VSS is in the pipeline; a preliminary study in nine patients showed a potential improvement [80] and an open-label treatment trial is ongoing [81]. A clinical trial using neurofeedback is also currently ongoing, with the idea of teaching patients to downregulate activity in different regions of the visual cortex (NeurofeebackSnow, NCT04902365). A study testing mindfulness-based cognitive therapy for VSS is also currently ongoing (NCT04184726).

The management of VSS comorbidities, such as migraine and depression, is paramount since it may reduce disease burden and improve coping behaviour. Cognitive behavioural therapy may be potentially useful since it is beneficial in depression and anxiety and is also used in the treatment of tinnitus. Patients can additionally be advised to join online patient groups such as the Visual Snow Initiative (www.visualsnowinitiative.org) or Eye On Vision Foundation (www.eyeonvision.com), as being part of a larger community that spreads awareness and information on the condition can often be helpful for sufferers.

Finally, it is interesting to note that novel treatments targeting the calcitonin gene related peptide appear ineffective both in VSS and visual aura, as reported in a recent case series [82].

## 5. Conclusions

Visual phenomena in migraine are common and sometimes of difficult diagnosis and management in clinical practice.

In this review, we gave a short clinical overview of the presentation of visual aura and visual snow, including what is known about their pathophysiology and management options. Both should be assessed in patients with visual symptoms, with or without associated headache.

There is a significant unmet need for scientific studies investigating these conditions; future research should particularly focus on treatment and management of patients, although in our opinion this is highly complex if the underlying pathophysiological mechanisms are not fully understood.

## Figures and Tables

**Figure 1 diagnostics-13-00252-f001:**
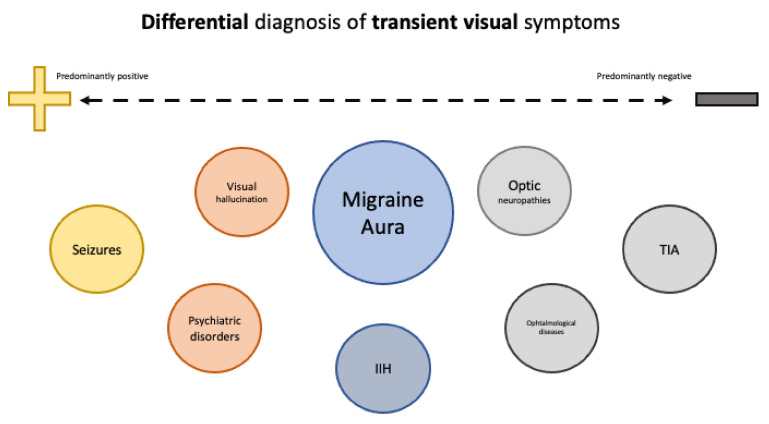
Differential diagnosis of transient visual symptoms, based on presence of predominantly positive vs. predominantly negative symptoms.

**Figure 2 diagnostics-13-00252-f002:**
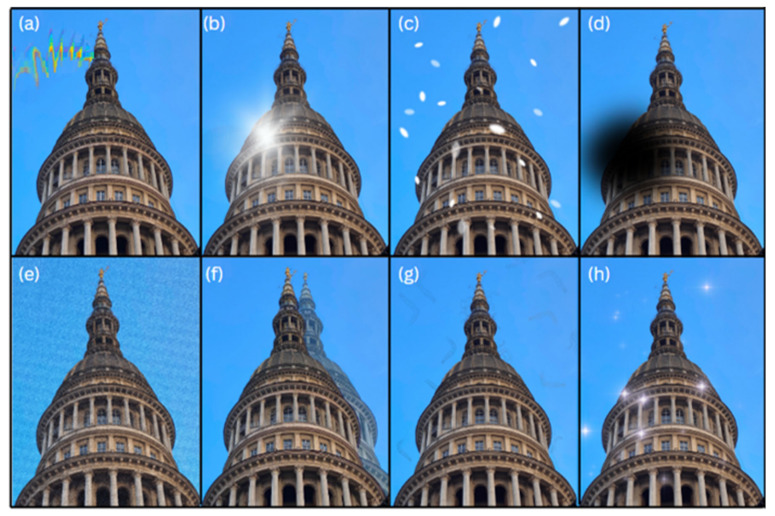
Typical symptoms of visual aura (**a**–**d**) and visual snow syndrome (**e**–**h**) showing similarities and differences among the two conditions: (**a**) zig-zag lines; (**b**) scintillations (**c**) white dots; (**d**) scotoma; (**e**) visual static; (**f**) afterimages; (**g**) floaters; (**h**) spontaneous photopsia.

**Figure 3 diagnostics-13-00252-f003:**
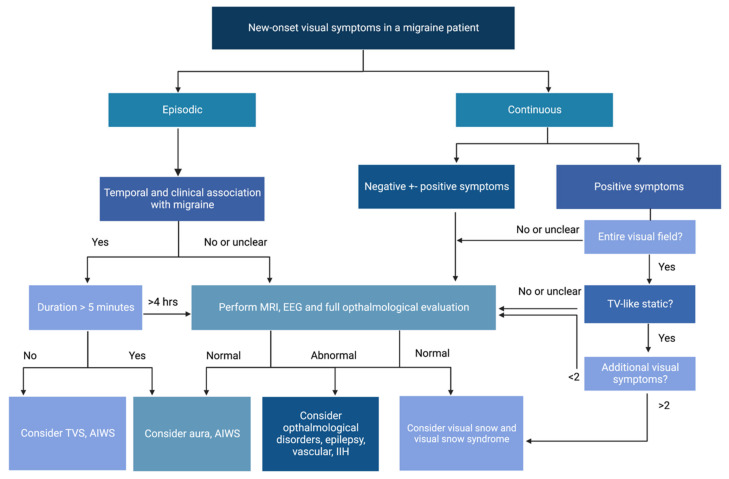
Diagnostic algorithm for visual symptoms of new onset in a patient with concomitant migraine.

**Table 3 diagnostics-13-00252-t003:** Possible secondary causes of visual snow syndrome.

Ophtalmologic
- Retinitis pigmentosa - Macular degeneration - Leber’s hereditary optic neuropathy - Autoimmune/paraneoplasic retinopathy - Rod-cone dystrophy - Bilateral neuropathies of toxic origin
**Neurologic**
- Idiopathic intracranial hypertension - Posterior visual pathway lesions (neoplasms, vascular or infections) - Multiple sclerosis - Creutzfeldt–Jakob disease (Heidenhain variant) or neurodegenerative processes - Epilepsy - Glycine receptor antibody syndrome
**Recreational drugs**
- Hallucinogen persisting perception disorder (HPPD)

## Data Availability

No new data were created or analyzed in this study. Data sharing is not applicable to this article.

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
