# Peer review of "Differential Diagnosis of Visual Phenomena Associated with Migraine: Spotlight on Aura and Visual Snow Syndrome"

_diagnostics, 2023, doi:10.3390/diagnostics13020252_

Round 1

Reviewer 1 Report

I read with great interest the paper entitled “Differential diagnosis of visual phenomena associated with migraine: spotlight on aura and visual snow syndrome” that explore the diagnostic and clinical differences among visual phenomena in migraine.

The review is well written and well organized. The topic is of great interest overall and for headache specialists and a review assessing differences in visual phenomena was lacking to date.

I have just some minor comments:

1.      Introduction (page 1, lines 44-48): please reformulate the sentences to better address that percentages refer to patients with aura and not the overall population with migraine.

2.     Treatment (page 7): although the focus of the review is on diagnosis, it could be useful to report and discuss the role of new treatments (anti-CGRP drugs [acute and preventive] and lasmiditan) on aura and visual snow and case report/study published.

3.     Conclusion (page 8): This paragraph should be extended, for instance adding the unmeet needs in diagnosis, type of studies and treatments.

 Typos:

1.     Please add a dot after table 1, table 2 etc. in table title.

Author Response

Point-by-point response to Reviewer 1

I read with great interest the paper entitled “Differential diagnosis of visual phenomena associated with migraine: spotlight on aura and visual snow syndrome” that explore the diagnostic and clinical differences among visual phenomena in migraine.

The review is well written and well organized. The topic is of great interest overall and for headache specialists and a review assessing differences in visual phenomena was lacking to date.

Response: We greatly appreciate the reviewer’s kind evaluation of our manuscript.

I have just some minor comments:

  1. Introduction (page 1, lines 44-48): please reformulate the sentences to better address that percentages refer to patients with aura and not the overall population with migraine.

Response: We thank the reviewer for this comment and have changed the sentence accordingly.

  1. Treatment (page 7): although the focus of the review is on diagnosis, it could be useful to report and discuss the role of new treatments (anti-CGRP drugs [acute and preventive] and lasmiditan) on aura and visual snow and case report/study published.

Response: We very much agree with this aspect and have added a paragraph on recent case series describing the effects of CGRP mABs on VSS and aura, in the final section of the discussion.

  1. Conclusion (page 8): This paragraph should be extended, for instance adding the unmeet needs in diagnosis, type of studies and treatments.

Response: We have expanded the conclusion as the reviewer suggests.

 Typos:

  1. Please add a dot after table 1, table 2 etc. in table title.

Response: We apologize for these typos, which have been amended.

Reviewer 2 Report

Authors realized a well done review about a very interesting subject that is not so studied in literature. Visual symptoms are very common in migraineurs, and burdening as well. The choice to consider together visual aura and visual snow syndrome (VSS) is partially justified: they are different conditions, but often coexist, and differential diagnosis between them is challenging. I believe that the paper will be very useful for medical community, both for research inputs and for clinical use.

I think that just a minor revision is needed.

As major concerns:  

- According to the article cited at reference (8), line 48, 0.01% is the esteemed prevalence of hemiplegic migraine, NOT of "motor symptoms"; please correct, clarify, or remove the information.

- At line 122, the most used term for CSD is cortical spreading "DEPRESSION", not depolarization. 

- VS is described most as a continuous symptom and sometimes episodic, how long do such episodes last? How long do VSS additional visual symptoms last?

As minor concerns: 

- Authors may consider to put a numeration to the paragraphs, unless differently stated by journal guidelines for a review.  

- Table 1 format should be improved. Some of the descriptions are a bit redundant (already easy to understand). Improve the formato of table 2 too. 

- a bigger picture showing a fortification spectrum could be nice. 

Author Response

Point-by-point response to Reviewer 2

Authors realized a well done review about a very interesting subject that is not so studied in literature. Visual symptoms are very common in migraineurs, and burdening as well. The choice to consider together visual aura and visual snow syndrome (VSS) is partially justified: they are different conditions, but often coexist, and differential diagnosis between them is challenging. I believe that the paper will be very useful for medical community, both for research inputs and for clinical use.

I think that just a minor revision is needed.

Response: We thank the reviewer for this kind and thorough assessment.

As major concerns:  

- According to the article cited at reference (8), line 48, 0.01% is the esteemed prevalence of hemiplegic migraine, NOT of "motor symptoms"; please correct, clarify, or remove the information.

Response: We apologize for this mistake and have rectified to ‘hemiplegic migraine’.

- At line 122, the most used term for CSD is cortical spreading "DEPRESSION", not depolarization. 

Response: We have now changed to ‘cortical spreading depression’.

- VS is described most as a continuous symptom and sometimes episodic, how long do such episodes last? How long do VSS additional visual symptoms last?

Response: We have added a description of length of episodes based on what has been reported in the VS literature. Unfortunately, reports on additional symptoms are less clear so we have omitted this aspect.

As minor concerns: 

- Authors may consider to put a numeration to the paragraphs, unless differently stated by journal guidelines for a review.  

Response: We have added numbers to paragraphs.

- Table 1 format should be improved. Some of the descriptions are a bit redundant (already easy to understand). Improve the formato of table 2 too. 

Response: We have changed Table 1 and Table 2 to best fit the valuable comment of the reviewer.

- a bigger picture showing a fortification spectrum could be nice. 

Response: We have changed the image to show a larger fortification spectrum.